# Ensuring Sustainability during a Crisis Using an Innovative Flexible Methodology

Daniela Dobreva Pastarmadzhieva, Mina Nikolaeva Angelova *, Stefan Atanasov Raychev, Blaga Petrova Madzhurova and Kiril Valkov Desev

Faculty of Economic and Social Sciences, University of Plovdiv Paisii Hilendarski, 4000 Plovdiv, Bulgaria; daniela.pastarmadzhieva@uni-plovdiv.bg (D.D.P.); raychev.stefan@uni-plovdiv.bg (S.A.R.); bmadzhurova@uni-plovdiv.bg (B.P.M.); kiril.desev@uni-plovdiv.bg (K.V.D.)
* Correspondence: mina.angelova@uni-plovdiv.bg

**Abstract:** The COVID-19 pandemic forced national governments and administrations to seek flexible solutions to deal with the emergency. Thus, the main purpose of the current study is to design a model of a flexible methodology based on detailed flexible methodologies to make decisions and measures connected to COVID-19 pandemic to be effectively applied without the loss of meaning and within a short time. For the creation of the methodology, we used comprehensive desk research based upon a literature review in the period May 2021–November 2021. As a result, an expandable set of relevant methodologies for crisis management and flexible methodologies was identified, modeled, and formalized using a broad literature review and an innovative model of a flexible methodology for crisis management was created in accordance with standardized concepts, transforming them into secondary use models. Furthermore, an algorithm for taking measures and decisions in crisis conditions was designed. The next step is to implement the methodology, which is planned for future empirical research. The findings provide an innovative model of a flexible methodology that could be used by academic and business representatives, public institutions of central and local government, and private stakeholders.

**Keywords:** crisis management; policy making; choice of technique; flexible methodology; COVID-19 pandemic

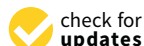

## 1. Introduction

Nowadays, crisis management is a key topic in research in different fields of sciences focused on pro-active suggestions to overcome the negative consequences on the economy, political environment, social environment, etc. In terms of economics and finance, the impact of the COVID-19 pandemic in Bulgaria on the labor market, economic activity, poverty, and inequality is growing steadily and is putting pressure on the government. What is happening can be defined as a constant search for a balance between the need to "restart" the national economy—to avoid social unrest and bankruptcies—and to prevent a consequent wave of the pandemic. This is a critical moment, because the danger of a slowdown in growth is the possibility of translating the economic crisis into the financial system. Thus, we need to identify the conditions that can limit the crisis to a subsystem.

The theoretical basis for crisis management, as part of the science of economics and management, has not yet found an integrative approach and has not developed and implemented a methodology to support the management of emergencies in societies (including those aimed at the so called "triangle of knowledge": science–business–society). The COVID-19 pandemic is an issue which causes novel problems and the existing solutions have proven to be insufficient. The countries are still struggling, and the indiscriminate change of government approaches is evidence that there is a niche to create new approaches or upgrade the existing ones.

The focus of the research is a set of flexible methodologies, namely, design thinking, user centricity, user innovation, agile, lean startup, and scrum. The subject of the paper is designing a model of a flexible methodology, which will enable the administration to take decisions and measures in emergency situations.

From the authors' standpoint, creating an innovative flexible methodology for crisis management contributes to better organization of measures and decisions connected with overcoming the COVID-19 pandemic consequences and can be implemented in future crises of similar type. To test our hypothesis, we used comprehensive desk research based upon literature review in the period May 2021–November 2021. The proposed methodology comprises four stages: (1) specifying an initial set of relevant methodologies for crisis management and project management; (2) modeling and formalizing flexible methodologies, including design thinking, user centricity, user innovation, agile, lean startup, and scrum; (3) generating an innovative model from standardized methodologies and developing a flexible methodology; (4) validating of the methodology and proving its significance to the practice.

As a result of the present fundamental research, a conceptual model and framework of the flexible methodology were established in the light of a specific problem, by analyzing and transforming it into a methodology based on expert political, economic, econometric, and social aspects. The proposed methodology is applicable to the specifics of crisis/emergency situations and can be adapted to the respective national context. This result should support the process of development and subsequent management of emergencies, including management of the crisis process and the overall management and implementation of measures to overcome its consequences.

The paper is structured as follows: following the introduction, the second part presents the methodology of the research. The third part is focused on the selection of methodologies, which are summarized based on a broad literature review. The latter is the framework of the study and is the basis for the fourth part, which presents the innovative methodology for crisis management. The paper closes with conclusions and recommendations for future research in the field of empirical implementation of the created own flexible methodology.

## 2. Methodology of the Research

The COVID-19 pandemic caused critical situations all over the world. The approaches of governments have varied across the globe and their common feature is the constant search for the right solution. The authorities have had to handle variety of problems and it seems the prioritization was not a strong point of the decision makers. As the right prioritization is a crucial issue in project management, the research team focused the efforts to examine a scope of methodologies for crisis management, project management and business process management. The members of the research team apply such methodologies in their practice and have preliminary knowledge and experience on the topic. The main purpose of the current study is to design a model of a flexible methodology based on detailed methodologies to make decisions and measures connected to COVID-19 pandemic to be effectively applied without loss of meaning and within a short time.

Many scientists aim to apply robust design methodologies focused on reducing the sensitivity of the design to variation. The robustness is typically evaluated using models, simulations or experiments and it is necessary to have in mind the possibility that the physical embodiment of the design might not satisfy the specifications due to uncertainties in the development process and lack of knowledge. The study of Roser and Kazmer [1] describes a flexible methodology aiming to minimize the expected cost of the design including development uncertainties.

For the purposes of the current research a four-steps methodology was used:

(1) Review of the literature specialized at project management methodologies. At this step the search key words were "project management methodologies" and "innovation methodologies". For scientific research were searched the following databases: Google Scholar, Research Gate, Academia, Science Direct, Publons. Furthermore, the team made a

Google search with the same keywords to find as much as possible websites, discussing the project management methodologies from practical perspective.

The research design method is deemed to employ project management methodologies and business process management methodologies as tools for investigating the research problem namely creating an innovative flexible methodology for crisis management. We carefully considered which methods are most appropriate and feasible for answering the research question. Aiming to explain the methodological procedures in such a way that any researcher can replicate, we stated as a base the research question: "Is there a flexible methodology to support the management of emergencies in societies?" The possible answers to this question are based on a literature review. The research design choice is driven by our aims and priorities, and it is based on qualitative research design that tend to be more flexible and inductive, allowing us to adjust our approach based on what we find throughout the research process. The type of qualitative design is the so called "Grounded Theory", aiming to develop a theory inductively by systematically analyzing qualitative data [2]. However, our preliminary study on the topic of COVID-19 pandemic and its influence toward the approaches of the governments [3,4] explore exactly what people struggle with in pandemic situation.

We found practical advises from many business consultants how to use different set of methodologies. Ben Aston states that: "In today's project management world, forward-thinking managers and leaders do not adhere to a single methodology- they become well-versed in many of them, and they learn how to mesh together various practices in order to accommodate whatever the project calls for." [5]. He presents methodologies and how to bring them into practice (i.e., agile, scrum, Kanban, scrumban, lean, eXtream Programming, waterfall, Prince2, PMI's PMBOK—Project Management Body of Knowledge).

We looked deep into different project management guides [6] and they reference approximately 8462 project management methodologies to choose from. The impact of the COVID-19 pandemic in Bulgaria on the labor market, economic activity, poverty, and inequality is growing steadily and is putting pressure on the government and, moreover, the main question is how the leaders will select which one is right for applying into the current situation. Our team focused the attention on data collection that could be useful in practical perspective. The next phase of the research methodology is data collection.

(2) Data collection is based on the sampling method and the available methodologies used into practical mode in research studies. We investigated our selected sample building inclusion and exclusion criteria to identify eligible studies. Because the data are very dense with information and ideas, we interpreted the meanings, identified the patterns, and extracted the parts that are most relevant to our research question. In the second step we used a comparative approach to select methodologies relevant to our general framework, namely, flexibility, applicability to high-risk projects and stakeholders' collaboration. The indicators are determined to be relevant to the COVID-19 crises but not to narrow too much the circle of options. The selection of methodologies to compare consists of adaptive project framework, agile, critical path method, design thinking, HAZOP, Kanban, lean startup, scrum, user centricity, user innovation, waterfall, and PMI's PMBOK. An initial examination of each of them was performed based on the specialized websites information and the following methodologies were selected as the most relevant: agile, design thinking, lean startup, scrum, user centricity, user innovation.

The preliminary study of the team was based on the SWOT (strengths, weaknesses, opportunities, threats) analysis. We conducted research on a variety of standards and processes in project management and crisis management integrated within the framework of the pandemic. The research of Ghorbani et al. [7] applies the analytic hierarchy process (AHP) and combined SWOT–AHP methods and, finally, created a new methodology called "strategy matrix" resulting from "priority matrix". They propose it to prioritize and determine the organization's possible strategies. We adopted the strategy matrix in the case of the pool of methodologies and their approbation in crisis conditions.

The research of Litt et al. [8] is interesting, and their focus to coordinate and share strategic approach that allows one to provide stability, coherence, and continuity to adaptation processes involving different stakeholders and sectors of the public administration. The methodology is based on established theories that draw on internationally agreed methodologies based on which the United Nations Framework Convention on Climate Change has developed a theoretical framework on adaptation. Their study helped us in establishing the criteria to include or exclude a study from the sample. We focused on the elaboration of a practical and specific methodology to support local, regional, and national decision-making bodies, structured in the following steps: build a knowledge base on methodology adaptation corresponding to crisis situations; assess the opportunities for crisis management and the methodologies' separately implementation; developing a complex flexible methodology consisting of the reviewed ones; explanation of the potential contribution of each methodology to the final model. We created a data management plan for organizing and storing our data based on the program Project Management Body of Knowledge. We applied discourse analysis focused on putting the data in context and involved analyzing different studies. We believe that finding a complex methodology to manage emergencies should be based on science. The preliminary list of different project management methodologies helped us to figure out which methods, principles, and approaches are appropriate to be used in a pandemic situation.

The scientific literature was the primary source of information. In addition to this, we used information from published articles by IT specialists and their experience in applying flexible methodologies, which is relevant to the issue investigated. Based on data gathering, analytical information was synthesized and produced for the purposes of the research. In addition, desk research was conducted, to countercheck the information against inconsistencies and to build correct and up-to-date logical model of reasoning. Table 1 presents the criteria for including methodologies in the sample.

**Table 1.** Criteria for including methodologies to the sample.

| Criteria for Inclusion | Examples |
|---|---|
| Practical implementation | Successful implementation in IT sector focusing on ICTs. Empirical studies that apply the methodology and gain successful results. |
| Stability | Optimization of processes and methods to minimize negative consequences. |
| Cohesiveness | This facet is envisioned to include the notions of collaboration, communication, and sharing, in addition to the notions of coordination and responsibility in the organization. |
| Validity | Specifics of crisis management and approbation of a complex methodology corresponding with the pandemic. |
| Software implementation | ICTs |
| Common Knowledge Base | The new flexible methodology can be able to manage crisis situations in the same domain of application. |
| Operationalization | Identifying the main methodologies 'concepts. Choosing features to represent each of the concepts. Selecting indicators for each of the features that are possible to be implemented into the new flexible methodology. |
| Robustness | The method is not affected when there is slight variation in the conditions. |

(3) The third step was the literature review. It helped us to go into details as concerns each of the methodologies and see how it was implemented in various situations. Thus, the strengths, weaknesses, opportunities, and threats of each methodology were identified, and it allowed to determine its contribution to the development of a flexible methodology for crisis management. Based on the literature review the team reached the conclusion that Scrum is not suitable for our purposes for the reasons discussed below.

Bialas et al. [9] present an advanced risk management methodology elaborated with a view to software implementation. It is a good example of ICT adoption in emerging domains of application. Their study tries to answer the question how to organize the revitalization decision process and support it with a software tool. Their methodology and tool are based on three pillars: risk, cost–benefit, and qualitative criteria assessments of the considered revitalization actions to select the target for the implementation of the given heap. Their findings focused our attention that the pool of methodologies must have the opportunity to be implemented in a software allowing the decision-makers to identify alternatives.

We reviewed previous studies to identify the most relevant or underused procedures and domains of the methodologies with a coverage to the new model creation. This highlighted any gaps in the existing literature that our research study filled. We selected some case study materials for the focus of our analysis grouped into four sections: methodologies applied in quantitative research (28 studies); methodologies applied in qualitative research (22 studies); methodologies applied in project management research (72 studies); methodologies applied in crisis management research (42 studies). The approbation of SWOT analyses gave us some critical insights about the final selection of methodologies that have the potential to be part of a new flexible methodology.

(4) The final step is the creation of a flexible methodology for crisis management, which incorporates elements of the final selection of five methodologies.

The final sample size of methodologies includes agile, design thinking, lean startup, user centricity, user innovation because of the initial examination of each of them and their capabilities to create a flexible methodology focused on:

- formulating solutions and testing hypotheses about the affected areas of social life by COVID-19;
- requiring non-traditional solutions;
- creating the citizen-led solutions platform as it should be constantly adapted to the changing environment, to the new realities and challenges;
- positive effects in boosting employee experience which in turn results in creating employee engagement;
- requiring novel decision and with the participation of representatives of all stakeholders.

The paper does not claim to be an exhaustive literature review about all listed methodologies and their practical implementation. A potential limitation was with this set of methodologies that the preliminary research based on the elaborated criteria excluded some of them as not applicable to the pandemic situation. As an important limitation of this finding was the fact that with a different set of indicators and/or with a different set of data, a different result may have been produced. This is because a unique solution for the provided paths is derived from the data (i.e., model identification) and thus, the results are data driven [10]. Future research would benefit from assessing the methodologies at multiple time intervals to establish test-retest reliability of the measures. Testing at multiple intervals would also help empirically support the expected stability of flexible methodologies noted in the literature.

## 3. Main Features of the Flexible Methodologies: Literature Review

Many authors focus their attention on the optimization of different processes and methods to minimize the negative consequences of the COVID-19 pandemic. The study of Pedrera-Jiménez et al. [11] sought to design and implement a flexible methodology based on detailed clinical models, which would enable electronic health records generated in a tertiary hospital to be effectively reused without loss of meaning and within a short time. Their methodology allowed the obtaining of the observation domain of the model with a coverage of over 85% of patients in most concepts. The authors furnished a solution to the difficulty of rapidly and efficiently obtaining electronic health records derived data for secondary use in COVID-19, capable of adapting to changes in data specifications and applicable to other organizations and other health conditions.

White and Marsh [12] focus on content analysis as a highly flexible research method that has been widely used in library and information studies (LIS) with varying research goals and objectives. It can be applied to many problems in information studies, either as a method by itself or in conjunction with other methods.

The research of Glas et al. [13] develop a flexible low-cost methodology for mapping flood hazard, vulnerability, and risk. A generic methodology was developed and customized for freely available data with global coverage, enabling risk assessment worldwide. It allows policy makers in developing countries to perform reliable flood risk assessments and generate the necessary maps.

### 3.1. Design Thinking

Design thinking is one of the nonlinear methods that can be used to deal with difficult problems that are undefined or unknown to understand the human needs involved, reformulating the problem, creating ideas through brainstorming, and by taking a practical approach to prototyping and testing [14]. Design thinking is an innovative thinking with orientation towards a radical innovation. It is based on the interdisciplinary principle [15].

Its origin dates to 1919, according to researchers, when the German architect Walter Gropius founded the School of Crafts, Design, Art, and Architecture. He began to use many elements that are part of the design thinking such as teamwork, removing hierarchies in the innovation process and reversing the design approach to user needs. After that, in the 1960s, collaboration in creative processes between designers, engineers and representatives of other disciplines became difficult, as there was often a difference in education and thus contradictory approaches to problem solving [16] and it became necessary to create a flexible methodology that eliminates these differences and systematizes the different ways of thinking.

The modern business approach to design thinking [17–20] considers it in a structured way. Brown's approach [17] or Ideo's conceptual design [18] show examples and outline the process of Design Thinking by not explicitly explaining the principles of design thinking. Brown [17] believes that so-called thinkers should be involved in the process from the very beginning. There should be a focus on the needs, preferences, and behavior of consumers, which make it possible to obtain new ideas and adapt them to the decisions. Experiments need to be carried out at an early stage and often enough to integrate external knowledge through users. Whether projects are short or long-term concepts need to be tracked, a certain budget and building of interdisciplinary teams are obligatory. To complete this goal staff from appropriate universities should be found. In addition, the process must be completed. Brown's model of procedure offers three phases [17]: inspiration, brainstorming, and performance.

Ideo's approach includes a questionnaire, structure, field research, expert interviews, storytelling, brainstorming [18]. The approach of Plattner, Meinel and Weinberg [21] is a summary of design thinking developed by the Institute in Potsdam and Stanford University. The principles used by Plattner et al. [21] include multidisciplinary teams as a necessity for the success of design thinking. The expert approach is diverse and does not consist only of specialists. Different learning styles are applied to encourage

thinking through open space with mobile and attractive furniture to encourage creativity and teamwork. The phases are understanding, monitoring, perspective, create ideas, prototypes, and test. The phases are interconnected and, therefore, feedback is possible to refine the findings and make corrections.

The method described by Liedtka and Ogilvie [20] involves an approach consisting of four phases and ten techniques. The phases are in the form of questions, namely:

- Phase 1: What is it;
- Phase 2: What if;
- Phase 3: What is wow;
- Phase 4: What works.

During these phases, divergent and convergent thinking are used in different ways. In the first two phases of the model the procedure is used divergently to break away from existing solutions. During the last two phases of the procedure model, the focus is then on the promising options, i.e., convergent.

Criticisms of design thinking should be noted. Buchanan [22] notes that the question of whose values matter and who should be involved in the design process has changed over time, evolving from the beliefs of the 1950s about "the ability of experts to create socially acceptable results" to the audience's view as "active participants in the conclusions" [22]. In today's environment, the distinction between designers and users is blurring, leading to the formation of a community of co-designers who fit their own context into the emerging design, thus constantly expanding it in different and undoubted ways. This orientation towards co-creation clearly introduces a social focus and emphasis on collaboration, which is lacking in earlier theories. Another element in today's views on design thinking is related to the role of empathy [23], a topic almost entirely absents from earlier theories. Empathy goes beyond simply acknowledging the subjectivity of the field of design. Virtually all current descriptions of the process emphasize design thinking as human-oriented and user-driven as a core value. In fact, Verganti [24] argues that the term "user-driven" is more appropriate to describe the approach than the popular term "design thinking". The "design-driven" strategy he formulates for innovation emphasizes the designer's ingenuity as a manager of choice, not a response to customer needs or demands. The addition is based on a strong design emphasis on the concrete and the visual to emphasize the key role of prototyping, which has long been a central feature in areas such as architecture and product development, but design thinking about prototyping differs from that of complex 3D prototypes and models traditionally found in these fields. The function of prototyping in design thinking is to stimulate real-world experiments in the service of learning, rather than showing, persuading, or selling [16].

The modern nature of design thinking was developed by Stanford University in 2010 and is called the "d.school's". The principles are the following [25]:

- Activity: Activity is more important than holding formal meetings.
- Experiment: Experiments and prototypes are part of the whole process and are used to gain experience.
- Empathy: Empathy for consumers is important for developing appropriate solutions.
- Visualization: Visualization is used to communicate ideas.
- Transparency: a common vision of different problems is formulated.
- Procedure: Techniques, objectives and position must be clear within the process.
- Diversification: Collaboration of team members from different backgrounds.

This is systematic thinking and is at the core of customers, business, and technology. The manufacturer brings together businesses (sales representatives and retailers), along with engineers and software experts and key customers. Based on this, they find a representative client who describes the needs in as much detail as possible and serves as a guide for this group of clients.

The essence of the model goes through five stages.

- Empathize involves understanding the problem to be solved. This includes consulting with experts, monitoring, engaging, and empathizing with people to clarify their experiences and motivations. Empathy is crucial to the human center of the design process, and empathy allows designers to separate themselves from their own assumptions about the world to gain an idea of consumers and their needs. Depending on the time constraints of this stage, a significant amount of information is collected to be used in the next stage [14].

- Define is the stage during which the information created during the previous phase is collected. Observations are analyzed and synthesized to determine the main problems that have been identified so far. This stage helps to gather ideas for identifying features and elements that allow to solve the problem or at least allow users to solve problems on their own with minimal difficulty [14].

- Ideate—during the third stage of the process of design thinking, designers are ready to start generating ideas. With the already available information, we start thinking non-standardly to identify new solutions and look for alternative ways to address the problem. The most used technique is brainstorming. It is important to get as many ideas or solutions to problems as possible at the beginning of the idea phase. Some other ideas techniques should be chosen by the end of the phase to help test so that the best way to solve the problem can be found [14].

- Prototype is the stage during which several cheap, scaled-down versions of a product or specific features found in a product are produced in order to be able to study the solutions to the problems generated in the previous stage. Prototypes can be shared and tested in the team itself, in other departments or in a small group of people outside the design team. This is an experimental phase, and the aim is to derive the best possible solution for each of the problems identified during the first three stages. The solutions are embedded in the prototypes and are tested one by one and are either accepted, improved, and reviewed or rejected based on user experience. By the end of this stage, the design team has a better idea of the limitations inherent in the product and the problems available, as well as how real consumers will behave, think, and feel when interacting with the final product [14].

- Test is the last phase. Designers or evaluators test the entire product using the best solutions identified during the prototyping phase. This is the last stage of the 5-stage model, but in an iterative process the results generated during the testing phase are often used to redefine one or more problems and inform users' understanding, conditions of use, how people think, behave, feel, and empathize. Even during this phase, changes and improvements are made to exclude solutions to problems and to obtain the deepest possible understanding of the product and its users [14].

The design thinking model is widely used in innovation processes. This methodology is at the heart of the success of Airbnb, Uber, Pill Pack, Clean Team, IBM, Stanford Hospital, Bank of America, and others. Modern solutions to problems would not be successful if this methodology was not applied. Furthermore, its capacity to be used for crises response has already been examined. Cankurtaran and Beverland [26] focus on the potential of design Thinking to support companies, especially B2B. The authors find that the method provide "rapid responses to emergent and fluid challenges" [16] (p. 259), which is the most needed solution in an unpredictable time such as the COVID-19 pandemic.

### 3.2. User Centricity and User Innovation

The concepts of user centricity and user innovation are tightly connected. To gain innovations from the users the organization must first adopt a user centric approach and then provide an environment for the users to innovate. Thus, first we need to discuss the very idea of user centricity and its implementation in the administration as an approach and after that to extrapolate the user-innovation concept into the flexible methodology for emergency situations management.

The involvement of the users in creating or enhancing some product or service is related to innovation. The user user-centric innovation concept became popular in the 1970s and nowadays it is very up to date topic. It means that the end consumer participates in the process of creation of an innovation and such approach proves to be an engine for improving the products or services [27].

However, it seems that the user centricity approach require focus, dedication, even a strategy so that the consumer is "at the heart of any development process". To achieve the latter the organizations, need to identify the "relevant people", to ask them the right questions and then to meet their expectations by taking the needed steps [28].

In fact, the effective and efficient engagement of the users in the process of creation or improvement of product or service is possible mostly because of the developments of the Information and Communication Technologies (ICTs) and the networking. In the context of COVID-19 pandemic the internet made it possible for most of the social structures to continue working. The administrations were forced to digitalize more of their services. Alongside, the disparities between various social groups regarding their access to ICTs become even more visible. In addition, it is not just the access to technologies but the ability to use them in accordance with own needs. These people are forced to use the new way of communication with the administration because of the social distancing requirements and at the same time this new way is inaccessible to them. There is a gap between the needs of the society and actual digital public services. The problem is rooted in the "administrative burden—a cognitive burden inflicted on citizens to make sense of a classification not based on their own lived experience but on the oftentimes opaque work organization of bureaucracies." [29] (pp. 1–2). Such an approach does not take into consideration the abilities of the users and their needs and thus, the e-governments cannot accomplish their goal [30]. We are convinced that the e-government is an area where the user centered approach cannot be regarded just as a good practice, but it must be the main manner of work. It can be also implemented in dynamic times such as emergency situations including COVID-19 pandemic. The user innovation approach provides tools for the realization of such an idea.

The user innovation as a concept gained publicity in the scientific research in the 1980s but it was not a brand new idea at that time [31]. According to Franke and Lüthje [32] (p. 2) "innovations by users represent a primeval and archetypical mode of innovation: if one has a problem, one tries to fix it" and currently, hundreds of millions of people are involved efficiently in user innovation processes.

The concept of user innovation is related mainly to the creation and improvement of products by the companies. It states that the consumers can also participate in this process, and this is not an area reserved only for the business [31]. The consumers produce innovations in accordance with their own needs for which the producer may not be aware at all and it is possible that many other users have same or similar expectations [33]. There are two main benefits for the latter from implementing such an approach—more sales and lower development costs [34]. Furthermore, the users cannot just offer improvements to the existing products, but they can also suggest the creation of a new product [35].

At some point the corporations have understood the importance of involving the users in the innovation process and some of them have created user innovation communities (UIC), "where customers submit their innovative ideas" [36] (p. 113). However, such a community may generate an enormous flow of information and the latter needs to be sorted and analyzed, and relevant conclusions must be produced. In this process of generating not only ideas but suggestions for actual innovations, the users contribute not only by the content, which they create, but also by their interactions [34]. Thus, those who create a UIC must be aware of how to manage the information flow and how to extract the relevant and significant suggestions.

*3.3. Agile*

Compared to traditional management approaches, Agile offers several key benefits to meet modern needs and a dynamic environment. On the one hand, it increases team productivity and employee satisfaction, reduces continuous repetitive planning and routine meetings, excessive documentation, reduces defects in product quality and characteristics. By improving accountability and continuously adapting to changing customer priorities, agile improves customer engagement and satisfaction, bringing desired products and features to market faster and more predictably while reducing risk. By involving individuals, specialists in many scientific fields as members of the team, the organizational experience is expanded and mutual trust and respect between the participants is built. Finally, by dramatically reducing the time wasted on micro-management of functional projects, it allows senior managers to concentrate on working with higher value and importance to the company [37]. The importance of using agile methodologies is to achieve higher quality software in a shorter period, based on self-organizing teams, cooperation with customers, less documentation and reduced time to market [38].

The starting point for the emergence of the agile approach is 2001, in which both the agile Alliance and the agile Manifesto were successively established. The manifesto sets out the whole philosophy of the agile management approach based on twelve basic principles and four core values. The principles are:

- The highest priority is to satisfy the customer through early and continuous delivery of valuable software. Customers are more satisfied when they receive working software at regular intervals, instead of waiting for long periods of time between versions.
- Welcome changing requirements, even late in the software development process. The agile methodology processes and uses these changes to achieve a competitive advantage for the customer.
- Providing working software as often as possible, within a few weeks to a few months, preferably in a shorter time.
- Businesspeople and developers must work together every day during the project.
- To build projects always around motivated individuals. To provide the environment and support they need on the one hand, and to trust them on the other so that they can do the job.
- The most effective and efficient method of transmitting information to and within the project development team is the face-to-face conversation.
- Working software is the main measure of project result and progress.
- Agile processes promote sustainable development. Sponsors, developers, and users must be able to maintain a constant pace of work indefinitely.
- Continuous attention to technical excellence and good design increases flexibility.
- Simplicity—the art of maximizing the amount of work not done—is essential.
- The best structures, requirements and designs arise from self-organizing teams.
- At regular intervals, the team should discuss how to become more effective, and then adjust and correct their behavior.
- The four core values corresponding to the principles set out in the manifesto and underlying the entire agile philosophy and methodology are:
- Individuals and interactions also take precedence over processes and tools—the agile methodology emphasizes that the people behind the processes are critical. It is the communication, cooperation and interaction between the team members that leads to solving the problems. The best tools in the wrong hands are completely useless.
- Running software in front of comprehensive documentation. Agile does not remove the documentation but optimizes it in a form that gives the working team what it needs to get started and build working software. Then, documents in the form of user reviews are what will help to improve future versions of the product.
- Cooperation of clients before negotiating the contract. According to agile's philosophy, the customer must be engaged and cooperate with the team during the development and creation process in the form of periodic demonstrations, his involvement as an

external member, part of the team who attends all meetings and ensures that the product meets his needs.

- Responding to change after the plan is implemented. The agile philosophy proposes to emphasize the need for change by moving from a static to a dynamic roadmap, which allows it to change from quarter to quarter, sometimes even from month to month.

Some of the most common and used methodologies within the philosophy of agile development. include extreme programming (XP), scrum, crystal, dynamic systems development method (DSDM), lean development, and feature-driven development (FDD) [39].

There are several challenges to the practical implementation of agile methods in the development of embedded systems leading to the conclusion that in the field of embedded product development there is a wide variety of products with different needs and specific problems, so no method is applicable, and many methods and practices are needed for different situations [40]. In their study, Laanti, Similä, and Abrahamsson [41], drawing on the many interpretations in the scientific and applied literature of agile philosophy, explore the extent to which they meet the principles and goals set forth in the Agile Manifesto. Psychological empowerment is empirically tested as an explanatory mechanism for the relationship between agile philosophy and the result. Empirically, the possibility of agile methods to be a means of empowerment and motivation for teams based on constructions for communication and team autonomy is validated [42]. Research is being done on the practical application and implementation of the agile philosophy in the real economy in view of the new economic realities, on the example of the Russian economic sectors [43]. The possibility of applying the agile philosophy not only on software development but also on the entire business management system is explored [44]. The future of business intelligent systems in the context of agile philosophy and the change that is taking place is studied [45]. A new approach in marketing based on the influence of agile philosophy and methodology—flexible marketing [46,47], and examines its application to the example of the tourism sector [48] or based on small firms in the real sector [49]. The relationship between the agile methodology and the health sector in the context of supply chains [50], health care [51] is investigated. In the context of the epidemiological situation and COVID-19, the possibilities of the agile methodology are explored [52,53].

The first one of the latter is focused on the opportunity to apply agile on national level and the second is focused on its application in a business organization in the new realities caused by COVID-19. Very interesting is the research of Janssen and van der Voort [52] focused on the potential of agile governance and the adaptive governance. The authors focus their research on the response of the Dutch government to the COVID-19 pandemic as a case study. The research give them evidence to conclude that there is "no single best response strategy", "responses may need to change over time", there is a need to "adapt, but ensure stability at the same time", "for adaptivity it is essential that government mobilize society" and "the value of having a variety of response strategies available" [52] (p. 6). Mancl and Fraser [53] (pp. 314–315) see agile as a tool to overcome the problems caused by the new reality to work from home. They list the challenges such as more responsibilities for the workers to supply themselves with equipment, isolation, overtime work, etc. The companies need to ensure "network-based digital collaboration tools" and agile is a method, which can be applied.

### 3.4. Lean Start-Up

The lean startup is a new approach being adopted across the globe, changing the way companies are built and new products are launched. It fosters companies that are both more capital efficient and that leverage human creativity more effectively. Lean focuses on people and teamwork at every level, in contrast to traditional management practices. In the late twentieth century, Peter Drucker [54] called for managers to act like scientists, and systematically and dispassionately investigate empirical evidence to detect threats and identify opportunities for new products and services. Blank and Dorf [55] state that the lean startup approach involves turning the underlying assumptions upon which a business

model is built into hypotheses that can be tested through the careful use of experiments. The promise of experimentation as an approach is that business model development can proceed faster, with higher certainty and lower resource requirements.

Hampel, Perkmann and Phillips [56] state that the idea of running experiments and then "pivoting" has become central to entrepreneurship practice on a global level and argue that entrepreneurship and innovation researchers should pay more attention to experimentation as an approach to innovation and corporate entrepreneurship in established firms.

There are lots of examples of successful enterprises that have rethought everything from governance and financial management to systems architecture and organizational culture in the pursuit of radically improved performance. Adopting lean relates to taking time and commitment, but it is vital for harnessing the cultural and technical forces that are accelerating the rate of innovation. While there is a growing body of research examining experimentation in startups, there is no corresponding literature investigating the role of experiments in the innovation and corporate entrepreneurship activities of established firms.

The core of the lean startup approach, based on Contigiani and Levinthal [57], is the rejection of an entrepreneurship model based on business planning in lieu of the adoption of an iterative approach driven by experimentation. The authors discuss the economic and technological forces that have caused the lean startup framework to have a natural saliency in the current business environment. Furthermore, they identify a variety of novel and interesting avenues for researchers in both entrepreneurship and the broader management literature that lie at the intersection of these domains. The provided summary of the lean startup framework and the existing perspectives of organizational leaning, real options, product development, and technology evolution, mixed perspective, with the design funnel suggesting the importance of an initial degree of parallelism with a subsequent emphasis on "building" and "testing" of a small subset of these efforts.

However, many authors, inspired by lessons from lean manufacturing, mark the importance that it relies on "validated learning", rapid scientific experimentation, as well as several counter-intuitive practices that shorten product development cycles, measure actual progress without resorting to vanity metrics, and learn what customers really want [58]. It enables a company to shift directions with agility, altering plans inch by inch, minute by minute.

An interesting research topic is the methodology of Lean Initiation and researchers analyze it as a process of management and development of enterprises, mainly states as dynamic. Villalobos-Rodríguez et al. [59] made contributions in this field and make brief comparative analysis with other similar methodologies in such a way that it is possible to objectively justify the use of this method.

The methodology helps organizations get out from stagnation and ensures sustainable growth. A lean startup has a rapidly changing business model which it uses to learn about its customers and about possible configurations of business components. Startup companies using the Lean approach also go through a sequence of phases, each of which emphasizes different aspects of the business model. Horton, Gors, and Knoll [60] propose a new solution that is motivated by specific attributes of the Lean Startup approach. The new business model architecture is intended to better support the startup process than previous designs.

Croll and Yoskovitz [61] give many examples and interviewed over a hundred founders, investors, intrapreneurs, and innovators proving the statement that "The Lean Startup movement is galvanizing a generation of entrepreneurs". Based on one of lean startup's core concepts namely build-measure-learn, the authors tackle some basic analytical concepts such as qualitative and quantitative data, vanity metrics, correlation, cohorts, segmentation, and leading indicators focused on providing a framework how to build something, measure its effects and learn from it to build something better the next time.

Many new methods were elaborated based on the Lean. For instance, the Lean Six Sigma is a method that relies on a collaborative team effort to improve performance

by systematically removing waste and reducing variation [62]. It combines lean manufacturing/lean enterprise and Six Sigma to eliminate the eight kinds of waste: defects, over-production, waiting, non-utilized talent, transportation, inventory, motion, and extra-processing. Six Sigma seeks to improve the quality of process outputs by identifying and removing the causes of defects (errors) and minimizing variability in (manufacturing and business) processes. Based on the statement of Bass and Lawton [63], lean aims to achieve continuous flow by tightening the linkages between process steps while Six Sigma focuses on reducing process variation (in all its forms) for the process steps thereby enabling a tightening of those linkages. Lean Six Sigma is used to reduce process defects and waste, and to provide a framework for overall organizational culture change.

### 3.5. Scrum

The increased competitiveness in the business environment and market globalization during the last decades, spur the invention and adoption of agile frameworks for more efficient product development. One of the most popular methodologies that is part of the agile approach is scrum [64]. The scrum emerged as a framework for new product development. It originates from a comprehensive study [65] in multinational companies from United States and Japan. The authors emphasize that there is arising need for speed and flexibility in the process of a new product development. Prior to the development of scrum, Takeuchi and Nonaka [65] conducted many interviews with employees and managers and found out that the leading companies had six characteristics in managing their new product development processes: built-in instability, self-organizing projects teams, overlapping development phases, multi-learning, subtle control, organizations transfer of learning.

A couple of years later the modern scrum was developed by the software practitioners Schwaber and Sutherland [66]. The framework is based on "The Scrum Guide" [66], where the following main components are described:

Scrum team: According to the framework, all the value is created through the fundamental unit of scrum which is a small team of people. The team consist of a scrum master, a product owner, and developers. There are typically 10 or fewer members of the team, which is small enough to be nimble and large enough to accomplish the goals of the sprint. The main advantages of the team are that it is self-managing and there are no hierarchies. The scrum master is responsible for the effectiveness of the scrum team, and as a leader, he supports the scrum team. The developers are the people in the scrum team who are responsible for the creation of the added value in each sprint. The product owner is responsible for maximizing the results from the scrum team's work. As so, he is accountable for the product backlog management, although he may delegate this work.

Scrum events. There are many events, described in the scrum framework, and all of them are design to provide the transparency required. The sprint is a container for all other events and all the work for achieving the product goal happen within Sprints. All activities for achieving a sub goal should be done intensively during time intervals (usually 2 weeks). At its core, each sprint is a short project, with defined tasks and outcomes. The rules in the scrum guide allow the cancellation of the sprint, but only by the product owner. The whole scrum team is responsible for the Sprint planning. The scrum guide defines 8 h time limit for a one-month sprint. Daily scrum is a 15 min meeting for the developers of the scrum team. The main purpose of this meeting is to produce an actionable plan for the next day. The focus of the daily scrum is the progress toward the sprint goal. A good practice during the meeting is for every participant to address what he did, what he will do the next day and what are the impediments for his work. Sprint review is held at the end of the sprint. The main topics that are discussed are: what was accomplished, what has changed in the environment and what to do next. The meeting should not be longer than 4 h and during it the scrum team presents the results of the sprint to key stakeholders. The progress towards the product goal is discussed as well. The focus of the Sprint Retrospective is to find ways to increase the quality and the effectiveness. The retrospective should take place, after the

last sprint, and is done by reflecting what went well and what did not. The retrospective time-box is maximum 3 h for a 1 month sprint. The scrum artifacts are product backlog, sprint backlog, increment.

The scrum guide elaborates on each item from the scrum framework insisting that it should be implemented in its entirety, otherwise it is not "Scrum". Although the scrum framework is widely used, there are evidence that sometimes it is more useful when some components of the framework are omitted. There are lots of proposals for hybrid approach that use only part of the scrum framework [64,67] or modifies its components [68] or sometimes deviates from scrum for sensible reasons [69].

Although scrum was intended for a new product development it shortly was widely adopted in the software development, and nowadays it is used in different areas such as science [70], finance [71], construction [72], public relationships [73], and education [74].

## 4. Flexible Methodology: Innovative Model Creation

Each of the flexible methodologies reviewed in the previous part can be implemented separately for crises management such as COVID-19. However, our team aimed at developing a complex flexible methodology consisting of the reviewed ones. Before presenting the model the potential contribution of each methodology to it is presented.

When formulating solutions and testing hypotheses about the affected areas of social life by COVID-19, design thinking could be applied with a certain budget. It should be noted that this is a methodology for collecting opposing hypotheses from all fields of science and would therefore, be a successful approach. This methodology is very suitable to be applied in emergency situations such as pandemics, because the latter are uncommon challenge and require non-traditional solutions. We believe that the first reaction needs to be design thinking as it allows free, creative, and intuitive thinking and it must be implemented in the very first days of the crises. Furthermore, such situations require teams consisting of experts from different areas and this is one of the solutions in the context of the situation. Regarding the pandemic, not only experts should participate but representatives of society and business. Such an approach can assure the basis of the design thinking, namely the empathy. Thus, the solutions will not be distant and bureaucratic but regarding the real problems of the stakeholders.

The user innovation approach can be implemented as a part of a flexible methodology for emergency situations management for several reasons. First, the emergency situations very often require non-traditional solutions. Different social groups face different challenges and sometimes the governments do not even have an idea what are the real problems. The COVID-19 crisis has increased the use of ICTs. Instead of trying to guess how to help people in a case of emergency, the authorities can build in advance an online platform for sharing problems and offering solution by the civil society.

Such platform can have several levels. On level 1, citizens can share their problems or share their ideas, opinions, solution propositions, etc., without both things being related. On level 2 people can react to the shared. As concerns the suggested ideas, the users can react to them by assessing them or suggest improvements. Regarding the problems of the citizens, the users may propose personal help or generate an idea how the authorities may solve such a situation or what measures should be adopted. On level 3 the users can react to the latter by assessing them or suggest improvements.

Of course, such a system, implemented on national level, may produce an enormous information flow, which needs to be sorted in some way. To be useful for the administration, the system must be able to extract the most popular and approved of them based on the suggestions and the reactions of the users. Thus, the personnel engaged with this process can analyze the more sorted and structured information. The analysts are the ones who must support the decision makers in taking the appropriate measures to deal with the crises, answering the variety of needs of the citizens. Furthermore, mutual assistance between citizens, which the system allows, can alleviate the state to some extent.

The agile approach can be implemented for creating a citizen-led solutions platform as it should be constantly adapted to the changing environment, to the new realities and challenges. The COVID-19 crisis has shown that in such a situation, the solutions must be provided very quickly, and any delay causes additional damages for physical health of the people, healthcare systems, business, economy, mental health of people, etc. Thus, agile dramatically reduces the time waste; it provides higher values in short terms. It allows the developers and the stakeholders to work together for the best solution. The agile team is crucial for the effectiveness of the measures against the pandemic; thus, the government should invest sufficient resources towards the motivation of the self-organizing team. There must be staff specially appointed for this work and it should not just be part of their other duties.

Moreover, the literature review focused on flexible methodologies presents many efforts to combine them. For instance, Blosch et al. [75] combine design thinking, lean startup and agile. Alongside with the potential positive effects the authors underline that such combination can lead to unintended results when executed by inexperienced executors. Furthermore, it can be used (un)intentionally to make a charade of the organizational transition it is supposed to support and become a waterfall in disguise.

When design thinking is applied to human resources, it has a positive effect in boosting employee experience which in turn results in creating employee engagement. Human-centric design thinking is a way of making work processes less complex and more enjoyable. The research of Durai et al. [76] intend to test the effect of design thinking, employee experience and employee engagement in lean startups. The findings highlight the importance of design thinking on employee experience that in turn will result in organic employee engagement.

The widely adoption of scrum framework (partly and as a whole) is evidence that the methodology could be very useful, when high quality and urgent results are needed. These are important requirements, especially in the field of crises management. There is a paucity of research [77,78] that aims to develop or refine such management frameworks during the COVID crisis. The scrum components that are particularly useful as building blocks of such a kind of framework are sprints, asynchronous daily scrum, and sprint retrospective.

To further improve the effectiveness of the aforementioned items of the scrum, software tools could be used. The cloud-based applications Zepel, Trello and Zoho offer the needed functionality to automate the components of the scrum framework.

Based on the examined methodologies, we suggest our own, which is presented in Figure 1.

We believe that in new emergency situations such as COVID-19 the design thinking approach is the best solution. The reason is because new situations require novel decision and with the participation of representatives of all stakeholders. The empathy is key to successful next step and, thus, this is Stage 1 in our methodology. To fulfill this step, user centricity and user innovation model is required, because this is the approach which is focused on considering the views and preferences of all stakeholders. To gather enough information and suggestions by the stakeholders, the authorities need to create an online platform which can enable the citizens and business to suggest anything related to the crises—both problems and solutions. Such a platform needs to be adaptive and to function throughout the whole pandemic, while considering new realities. For the creation and maintenance of the platform, the agile methodology is suitable. Agile provides an algorithm for constant adaptation of the software and this is very valuable in crises. Although scrum has many advantages, it has too many formalities and we believe that it is not very suitable for the situation, which requires dynamics and more freedom in the decisions.

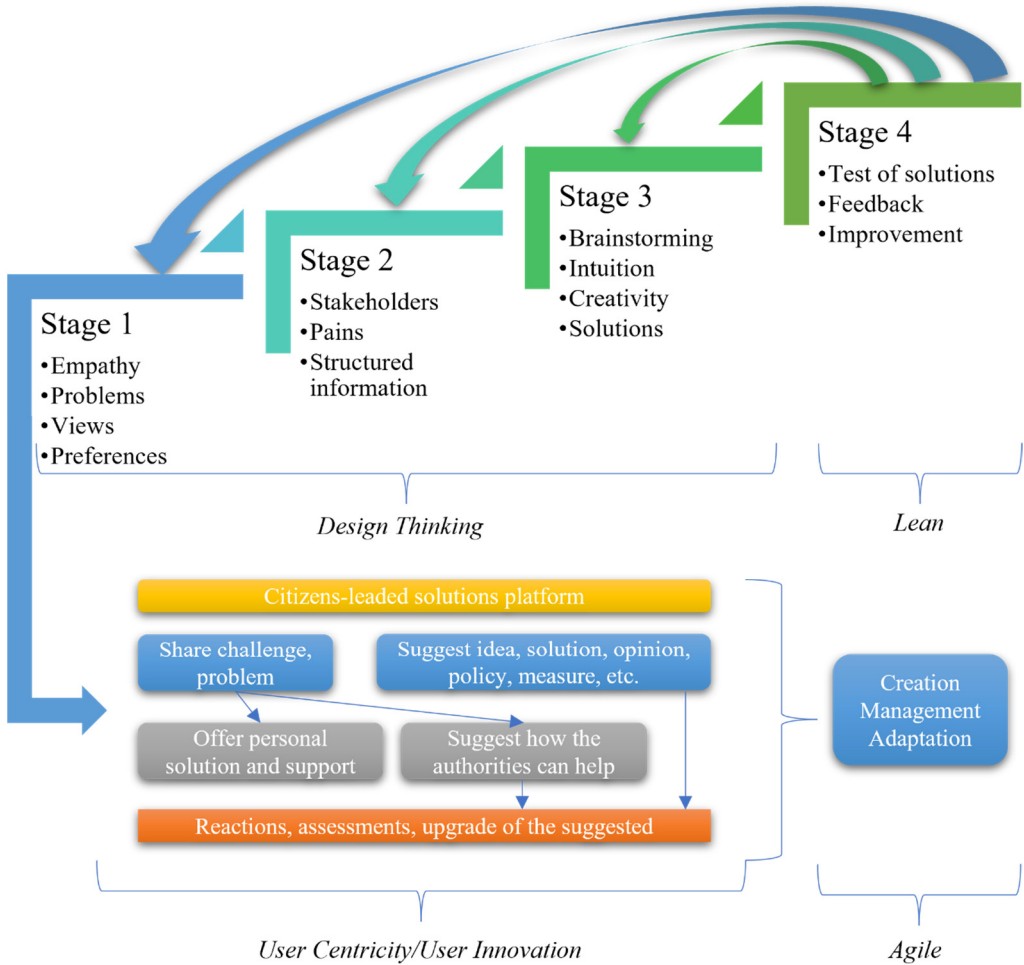

**Figure 1.** Flexible methodology for emergency situations.

After the acquisition of data from the platform, Stage 2 must be performed. During Stage 2, the different stakeholders must be described, together with their pains. This step will provide a detailed picture of the variety of challenges faced by different social groups, and it must be performed very precisely. This group must perform more administrative—rather than creative—work. They need to summarize, in the most detailed possible way, the data accumulated in the platform and present it in a structured way for Stage 3 of the process. As Daraio et al. [79] state: "in the Big Data era, a crucial role is played by data quality".

Stage 3 is essential. Ideate is the phase where the brainstorming takes place. The participants need to have the most detailed information prepared in Stage 2. However, they must be invited to be as intuitive and creative as possible during the brainstorming. The group must consist of experts from different areas, from which areas need be identified from the analysis in Stage 2. The result of Stage 3 must be a list of ideas for short, mid, and long-term solutions. On the one hand, it should suggest how to support the different stakeholders while the crises are ongoing, and on the other it should provide ideas on overcoming the crises.

Stage 4 is inspired by lean methodology and is related to its focus on hypothesis test through experiment. Lean can be implemented in pandemics as the model of reactions, taking measures and interactions with the business and citizens must be constantly adapted and improved. In such a situation, the authorities need to act rapidly in implementing solutions, measuring their effect, learning from mistakes, and upgrading the solutions. The reasons for unsuccessful solutions need to be identified rapidly and the right solution needs to be found.

When a stage is reached, the previous does not stop working. In the core of the presented methodology is the belief that the dynamics of pandemics are to a high extent unpredictable, and a constantly adapting and creative model is required.

## 5. Conclusions

The COVID-19 emergency has put countries' governments in a brand-new situation. Some of them have succeeded in finding solutions and have kept the pandemic under control. This saved lives and protected the healthcare system and economy. Others did not manage to deal very well with the crises. The causes for success and failures can be different, but we are convinced that the COVID-19 pandemic demonstrated that governments must be ready for flexible solutions and that is why we turned to some flexible methodologies. We aimed at identifying whether they can be useful in creating strategies for reaction in emergencies.

The emergencies, especially when they are new, do not need a ready solution burdened with formalities. This may cause more problems and more challenges for the system. We believe that the suggested model can be very useful because it considers the problems of different social groups, develops an environment for creative solutions, and provides opportunities for these solutions to be constantly upgraded. The empathy in the model is the prevention of distrust in the government, which seems to be one of the most significant problems in the pandemic.

**Author Contributions:** Conceptualization, M.N.A. and D.D.P.; methodology, M.N.A. and D.D.P.; validation, M.N.A. and D.D.P.; formal analysis, M.N.A. and D.D.P.; investigation, M.N.A. and D.D.P.; resources, M.N.A., D.D.P., S.A.R., B.P.M. and K.V.D.; data curation, M.N.A., D.D.P., S.A.R., B.P.M. and K.V.D.; writing—original draft preparation, M.N.A., D.D.P., S.A.R., B.P.M. and K.V.D.; writing—review and editing, M.N.A. and D.D.P.; visualization, D.D.P.; supervision, M.N.A. and D.D.P.; project administration, M.N.A.; funding acquisition, M.N.A. All authors have read and agreed to the published version of the manuscript.

**Funding:** The paper is part of a project No KP-06-DK-2/7/2021, funded by Bulgarian National Science Fund.

**Institutional Review Board Statement:** Not applicable.

**Informed Consent Statement:** Not applicable.

**Data Availability Statement:** Not applicable.

**Conflicts of Interest:** The authors declare no conflict of interest.

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
