# Peer review of "Ensuring Sustainability during a Crisis Using an Innovative Flexible Methodology"

_sustainability, doi:10.3390/su14052996_

Round 1

Reviewer 1 Report

Review Manuscript Sustainability-1596364

Dear author(s),

First of all, thank you for the opportunity to review this manuscript. Please, see below some suggestions for your evaluation if they can contribute to improve your work.

Please consider defining what is meant by "flexible methodology", as this is central to the paper, so better conceptual precision is advisable.

How was the group of methodologies of interest defined (i.e., Design thinking, 50 User Centricity, User innovation, Agile, Lean Start-up and Scrum.)? Why (which criteria) were these methodologies chosen and not others? This is important to know as it contributes to the understanding of the methodological procedures that were used by the authors to select (or eliminate) the studies to be reviewed.

Please consider in the introduction to discuss the topic and its importance in relation to the relevant literature, demonstrating the existence of a theoretical and/or methodological gap.

Please strongly consider developing a methodology section. It is stated that the paper consists of a literature review, but the methodological procedures followed are not explained. What are the search keywords? What bases were considered and why? What are the criteria to include or exclude a study from the sample? What types of studies were considered? What are the intermediate steps (if any) for a detailed evaluation of the studies to be included? How was their relevance determined? Was some type of scientific methodological framework such as PRISMA used? What is the final sample size? These and other questions need to be answered so that the rigor of the study can be attested. They directly influence the results and contributions of the paper, as they are based on your sample and the way in which it was arrived at.

Line 67 - Please check the spelling of the following sentence: "The proposed is applicable and adapted to the specifics of…"

Line 69 – Please consider not abbreviating words like "incl."

Line 81 – Please check the spelling of “Pedrera-Jim´enez et al”

Line 89 - Please check the proper form of citation in the text for “White & Marsh”

Line 93 - Please check the punctuation in this sentence

Line 103 – Please consider explain better what “radical orientation” means

Line 104 – This sentence requires reference(s)

Please consider standardizing the writing of terms throughout the text (e.g., Design Thinking, design thinking)

Lines 136-137 – It is necessary to improve this sentence (use of capital letter, colon in a row, punctuation, etc.)

Were there no studies that relate Design Thinking and covid-19?

Line 244 – Please state what “ICT” means in its first use in the text

Line 253 - To affirm this, it is necessary to explain how this statement was arrived at (or conviction, as used in the text), on which references/data this statement is based.

Lines 260-261 - The quoted sentence requires the page

Lines 262-264 – There is a need to explain how this statement was arrived at. The readers may ask why the authors find it relevant besides their opinion? Or how to base this opinion with data, arguments and scientific references?

Line 275 - The quoted sentence requires the page

Lines 283-291 - The statements made require references

Lines 369-371 - Considering that the covid-19 pandemic is a central factor in the paper, wouldn't it be valuable to discuss references 41 and 42, precisely those that relate the methodology to covid-19?

Line 384 - Please check the proper form of citation in the text for “Hampel, Perkmann & Phillips”

Line 397 - Please check the proper form of citation in the text for “Contigiani & Levinthal”

Line 427 - Please check the proper form of citation in the text for “Croll & Yoskovitz”

Line 454 – Missing reference for Takeuchi and Nonaka

Line 573 – Missing reference for Zepel, Trello and Zoho

Please consider using “Figure 1” instead of “Scheme I”

Line 583 - Please check whether it is necessary to capitalize the word “First”

Line 594 - Please check whether it is necessary to capitalize the word “Second”

Line 602 - Please check whether it is necessary to capitalize the word “Third”

Line 610 - Please check whether it is necessary to capitalize the word “Fourth”

Please consider standardizing the use of terms (e.g., stage 1) as this can facilitate reader understanding by associating text with figure

Thanks again for the opportunity to review this work.

Author Response

Dear reviewer,

Thank you for your valuable comments and recommendations. We tried to apply all the suggestions for improvement, and we hope that the paper became much more interesting for the reader. Your efforts and precise review proved the quality of the final paper! The paper is in “track changes” function, such that changes can be easily viewed by you.

Reviewer 2 Report

Because this is a review of design management approaches hypothetically applicable to the management of Covid-19 based crises, the empirical evidence to support the authors conclusions is (necessarily) rather limited. Given that the authors are - at least in part - discussing the design of management systems to be robust to unexpected challenges, I frankly think the assessment would benefit from looking at engineering design approaches including (for instance) HAZOP. Yet, this manuscript is valuable. 

Author Response

Dear reviewer,

Thank you for your valuable comments and recommendations. Your suggestions are discussed in the new section 2. Methodology of the research. 

We explained in detail the steps that the methodology goes through. The selection of methodologies to compare consists of Adaptive Project Framework, Agile, Critical Path Method, Design Thinking, HAZOP, Kanban, Lean Start-up, Scrum, User Centricity, User Innovation, Waterfall. An initial examination of each of them was performed based on the specialized websites information and the following methodologies were selected as the most relevant: Agile, Design Thinking, Lean Start-up, Scrum, User Centricity, User Innovation.

The paper is in “track changes” function, such that changes can be easily viewed by you.

Thank you for the suggestions for improvement, and we hope that the paper became much more interesting for the reader.

Reviewer 3 Report

A very interesting paper that addresses a very relevant and timely topic. I found the research to be original and well presented, and explained in a clear

Author Response

Dear reviewer,

Thank you for your valuable comments and appreciation you have given to our paper.

Round 2

Reviewer 1 Report

Dear author(s),

Thank you for the opportunity to review the improved version of your work.

Minor issues were addressed, but the main one (methodology), which affects the core of the study's quality, still contains issues that need to be improved. Please bare in mind that it is necessary to explain the methodological procedures in such a way that any researcher can replicate. Although this new version has a methodology section, the description is not enough. You explained what you did, but you didn't explain how. Both are crucial for a methodologically rigorous study. What are the criteria to include or exclude a study from the sample? What types of studies were considered? What is the final sample size? At least these questions need to be answered. Without this, it is not possible to know which sample generated the analyzes and results. Without a properly explained methodology, this text cannot be classified as a "research paper", as it is not replicable and it is not possible to assess the methodological rigor. I strongly recommend researching literature review studies in the journal Sustainability so that they serve as a reference for quality and methodological rigor.

Also, the text needs proofreading as there are many typos throughout it. I point out a few below, but you need to check the entire text.

Lines 72-73 - The proposed methodology is applicable and adaptapplicableed to the

Line 88 -  the prioritization was notn’t a strong point

Line 94 - Review of the literature and web basedweb-based platforms

Please check the entire text to clarify issues such as those noted below:

Line 94 - What web-based platforms specialized at project management methodologies were used?

Lines 97-98 - Indicate only the name of the data bases, not the websites

Lines 98-99 – “We also referred to resources, previously available to the team” – What does that mean? It is not possible to understand.Thanks again for the opportunity to contribute to your work.

Author Response

Dear Reviewer,

Thank you for your comments concerning our manuscript entitled “Ensuring sustainability during a crisis using an innovative flexible methodology”! They are all valuable to us and were very helpful for revising and improving our paper. We have studied your comments very carefully and have made corrections according to your suggestions. We hope to meet them with approval. The paper is in “track changes” function, such that changes can be easily viewed by you.

Round 3

Reviewer 1 Report

Dear authors,

Thank you very much for this new improved version of the paper.

Please check the attached file for you to understand some comments where I was not clear enough. In this file there are three images:

Image 1: Please note how the text gets confused, mixing added and deleted words/sentences, not being possible to differentiate what continues or not in the paper (including the colors of the "track changes"). Perhaps this is a technical problem of the system when converting the file, however, this does not take away the fact that it makes it difficult to read and, consequently, the work of the reviewer.

Image 2: Please note that, in addition to what was mentioned for the first image, in this one it is clear how difficult it is to know what is or is not in the text. For example, I had to assume that "[???]" is not, as are the links (which, in the only file I have access to, appear mixed with the names of the databases).

Image 3: Please notice in this figure how the reviewer's job becomes extremely difficult than it is supposed to be. I am obliged to believe that only one of the tables will be considered, and the only thing that indicates this is "???", as the colors are the same and there is no indicative. In addition, this impoverishes the look of the article, as well as the fluidity of reading.

Regarding the recommended English proofreading of the text, here is an example of why I believe this is necessary, even by someone fluent:

Lines 147-148: "We recruit our selected sample building inclusion and exclusion criteria to identify eligible studies" - Is saying that the studies were recruited the most appropriate way of saying it? Especially in the area of management, to whom I believe this text may be of interest, the term "to recruit" does not seem the best. The entire text needs to be verified accordingly.

Author Response

Dear Reviewer,

Thank you for your comments concerning our manuscript entitled “Ensuring sustainability during a crisis using an innovative flexible methodology”! They are all valuable to us and were very helpful for revising and improving our paper. We have studied your comments very carefully and have made corrections according to your suggestions. We hope to meet them with approval.

According to the journal instructions, we prepared the paper in “track changes” function, such that changes can be easily viewed by you. But perhaps there is a technical problem of the system when converting the file. Based on your images I understood that you do not use the version of the manuscript that I uploaded. I would like to apologize for the inconvenience, but this issue is a technical problem of the system.

The current paper is not in “track changes” function and I marked all the corrections in red.

Lines 147-148: Regarding the recommended English proofreading of the text, we applied for the services of the journal.